# Electrospinning and Rheological Characterization of Polyethylene Terephthalate and Polyvinyl Alcohol with Different Degrees of Hydrolysis Incorporating Molecularly Imprinted Polymers

**DOI:** 10.3390/polym16233297

**Published:** 2024-11-26

**Authors:** Sisonke Sigonya, Teboho Clement Mokhena, Paul Mayer, Talent Raymond Makhanya, Thabang Hendrica Mokhothu

**Affiliations:** 1Department of Chemistry, Durban University of Technology, P.O Box 1334, Durban 4000, South Africa; talentm@dut.ac.za (T.R.M.); thabangm1@dut.ac.za (T.H.M.); 2DST/Mintek NIC, Advanced Materials Division, Mintek, 200 Malibongwe Drive, Randburg 2194, South Africa; tebohom@mintek.co.za; 3Department of Chemistry and Biomolecular Sciences, University of Ottawa, 150 Louis-Pasteur Pvt, Ottawa, ON K1N 6N5, Canada; paulmichael.mayer@uottawa.ca

**Keywords:** electrospinning, polyethylene terephthalate (PET), polyvinyl alcohol (PVA), molecularly imprinted polymers (MIPs), rheological properties

## Abstract

This study investigates the electrospinning and rheological properties of polyethylene terephthalate (PET) and polyvinyl alcohol (PVA) with varying degrees of hydrolysis (DH) for molecularly imprinted polymer (MIP) incorporation. The morphology and properties of the electrospun nanofibers were evaluated, revealing that PVA nanofibers exhibited smoother and more uniform structures compared to PET fibers. The rheological behavior of the polymer solutions was also characterized, showing that PVA 99 DH solution exhibited shear-thinning behavior due to the unique structural properties of the polymer chains. The introduction of MIP and NIP additives had no significant impact on the rheological properties, except for PVA 99 MIP and NIP solutions, which showed deviations from Newtonian behavior. The electrospun MIP nanofibers showed a conductivity of 1054 µS/cm for PVA (87–90% DH) and a viscosity of 165.5 mPa·s, leading to optimal fiber formation, while displaying a good adsorption capacity of 0.36 mg for PVA-MIP to effectively target pharmaceuticals such as emtricitabine and tenofovir disoproxil, showing their potential for advanced water treatment applications. The results suggest that the electrospinning process and rheological properties of the polymer solutions are influenced by the molecular structure and interactions within the polymer matrix, which can be exploited to tailor the properties of MIPs for specific applications.

## 1. Introduction

The presence of pharmaceutical compounds in aquatic systems is a growing global environmental issue. Among these pollutants, antiretroviral drugs (ARVs) and non-steroidal anti-inflammatory drugs (NSAIDs) are particularly problematic due to their widespread usage and persistence in wastewater [1,2,3]. Traditional wastewater treatment methods often fall short of effectively removing these compounds, necessitating innovative approaches for their adsorption and removal from water systems [4]. Electrospun polymeric materials, enhanced by molecularly imprinted polymers (MIPs), have emerged as promising candidates for the selective adsorption of pharmaceutical pollutants, leveraging their high surface area, tunable properties, and tailored recognition sites [5,6,7].

Molecular imprinting represents an innovative method for the selective recognition and binding of pharmaceutical contaminants due to their tailor-made cavities, which are complementary in size, shape, and functional groups to the target molecule [8]. Despite their specificity, MIPs alone cannot always cope with the complex matrix of wastewater. Thus, to enhance their practical application, it is efficacious to incorporate MIPs into functional materials that facilitate their deployment in water treatment systems.

Electrospinning is a versatile technique for producing nanofibers with exceptionally high surface area-to-volume ratios [9,10]. Incorporating MIPs into electrospun polymer matrices, such as polyethylene terephthalate (PET) and polyvinyl alcohol (PVA), can significantly enhance the adsorption capacity and selectivity of these materials for specific pharmaceutical pollutants. The selection of PVA and PET as primary polymers for the electrospinning process is grounded in their advantageous properties for water treatment applications. PVA is recognized for its biocompatibility and aqueous solubility, which facilitate the formation of uniform nanofibers essential for effective filtration membranes that selectively permit water passage while retaining contaminants [11,12]. Meanwhile, PET offers significant mechanical strength, thermal stability, and chemical resistance, making it suitable for constructing durable membranes capable of enduring harsh environmental conditions [13,14].

The use of these polymers enhances the structural integrity and performance of the membranes, allowing for property tailoring to optimize filtration efficiency and durability. Existing literature supports the efficacy of these polymers in electrospun membrane applications for water treatment [13,15,16]. MIPs are synthetic polymers designed with specific binding sites that can selectively capture target molecules, making them ideal for applications where specificity is crucial, such as the removal of ARVs and NSAIDs from wastewater; various studies have been published showing the effectiveness of synthesized MIPs [17,18,19,20,21].

This study investigates the synthesis and characterization of electrospun solutions composed of PET and PVA solutions with MIPs. Fourier Transform Infrared (FTIR) spectroscopy was used to explore the chemical interactions and functional groups within the materials. Key physical properties such as viscosity and conductivity were also assessed to evaluate their effects on electrospinning efficiency and fiber morphology.

Thermogravimetric analysis (TGA) was conducted to determine the thermal stability of the synthesized materials, a crucial factor for their application in various environmental conditions. Additionally, chromatographic techniques were utilized to analyze the adsorption performance, focusing on uptake capacity and kinetics. To assess the morphology and structure of the electrospun fibers, scanning electron microscopy (SEM) was utilized, providing insights into fiber diameters and surface characteristics vital for optimizing electrospinning parameters. This interplay between electrospinning conditions and the resultant nanofiber properties is crucial for refining their practical applications in adsorption.

Ultimately, this research aims to advance the understanding of electrospun MIP-composite membranes for the effective removal of pharmaceutical pollutants, addressing the significant challenge of water contamination by persistent substances. Insights gained into the structural, thermal, and adsorption characteristics of these materials will facilitate the future development and optimization of advanced adsorption technologies. This study includes the development of a new class of filtration media that can specifically target and remove a wide array of pharmaceutical compounds from wastewater with high efficiency. We aim to mitigate the environmental impact of these persistent pollutants and address the public health concerns associated with their presence in various water supplies.

This study provides novel insights into the synthesis and characterization of electro-spun MIP composite membranes utilizing PVA and PET, in addition to the innovative application of molecular imprinting technology in tailoring polymer fibers that exhibit better selective adsorption capabilities specifically targeting persistent pharmaceutical pollutants, such ARVs and NSAIDs. The results show that electrospun MIP composites not only enhance adsorption performance through the formation of specialized binding sites but also demonstrate important rheological properties influenced by polymer characteristics and processing parameters. Furthermore, the development and application of these membranes provide a tangible solution to the limitations of current detection and quantification methods. While traditional analytical techniques offer reliable data, they are not without drawbacks. Sample preparation can be laborious and time-consuming, potentially introducing errors or losses of target analytes.

## 2. Materials and Methods 

### 2.1. Materials and Instrumentation

Acrylamide, 1,10-azobis-(cyclohexacarbonitrile), ethylene glycol dimethacrylate (EGDMA), acetonitrile, polyethylene terephthalate (PET), polyvinyl alcohol (PVA), emtricitabine (EMI), tenofovir disoproxil (TENO), naproxen (NAP), diclofenac (DICLO), ibuprofen (IBU), and efavirenz (EFV) ± 98%, respectively, were obtained from Sigma-Aldrich, Canada Ltd. (Oakville, ON, Canada) and used as received. All solvents were sourced from Merck, Canada, and utilized without further purification. Aqueous solutions were prepared using purified water.

Scanning electron microscopy (SEM) was conducted utilizing a JEOL JSM-7500F (Tokyo, Japan) field emission scanning electron microscope. For ultra-high-performance liquid chromatography (UPLC) analyses coupled with UV–VIS detection (LC-UV), the chromatographic separation and identification of non-steroidal anti-inflammatory drugs (NSAIDs) and antiretroviral drugs (ARVs) were performed.

### 2.2. Synthesis and Electrospinning Membrane

#### 2.2.1. Preparation of Molecularly Imprinted Polymer Microsphere

MIP microspheres were synthesized using a two-phase MIP bulk polymerization process. In the first phase, a mixture of 1,10-azobis-(cyclohexanecarbonitrile) and ethylene glycol dimethylacrylate was reacted in toluene under nitrogen at 70 °C for 8 h. In the second phase, a solution of template molecules (EMI, TENO, NAP, DICLO) in acetonitrile was combined with additional reagents and added to the reaction product from the first phase. The templates were removed using a Soxhlet extraction process. The resulting polymer was washed repeatedly with a 10% (*v*/*v*) mixture of acetic acid in methanol until the templates were no longer detected by the LC system. Following this, the polymer was dried thoroughly, and MIP particles were then sized and separated for extraction experiments and characterization.

#### 2.2.2. Preparation of Molecularly Imprinted Membrane (MIM) Via Electrospinning

PET solution was prepared by mixing 15% of PET with 4.0 mL of TFA in sealed vials and stirring for 3 h at room temperature. Subsequently, a solution of molecularly imprinted polymers in 30 mL dichloromethane (DCM) was added to the PET solution. After 2 h of agitation and ultrasonication, the same procedure was followed for PVA with a DH of 87–90% and PVA with a DH of 99%, where 7 g of each polymer was dissolved in 40 mL of different solvents including WATER (40 mL), ethanol (10 mL) and water (30 mL). The resulting solutions were then loaded into a plastic syringe connected to a syringe pump (KDS-200, Focus Co., Ltd., USA, Paramus, NY, USA). The positive electrode of high-voltage power supply was attached to the metal needle tip (0.8 mm) of the syringe. The metal collector was covered with aluminium foil as a collector. An 18 KV voltage was applied with a tip-to-collector distance of 15 cm. The ambient conditions were controlled at a temperature of 20 °C and a relative humidity of 40%. The fibrous membranes were then dried in a vacuum at room temperature to eliminate any residual solvent. NIMs were synthesized using the same method but without templates. The morphology and size of the polymer microspheres and electrospun membranes were characterized using scanning electron microscopy (Figure 1).

### 2.3. Analytical Conditions

#### 2.3.1. Optimization of MIP Amount

Molecularly imprinted polymer membranes of 10.0 mg, 20.0 mg, 30.0 mg, 40.0 mg, 50.0 mg and 60.0 mg were added into an aqueous solution containing EMI, TENO, NAP and DICLO (40 μmol/L, 10 mL). Then, the samples were detected and analyzed after continuous stirring for 2 h. The solution of 20 μL was removed for the analysis with HPLC each time.

#### 2.3.2. Optimization of Adsorption Time

Optimization of adsorption time is important for maximizing adsorption efficiency and capacity in various applications. Carefully adjusting the duration of contact between the composite material and the target substances enhances the kinetics of adsorption, improves pollutant removal rates, and ensures the effective utilization of adsorption sites. Understanding the optimal adsorption time for PVA-MIP composites allows for tailored design of adsorption systems, leading to increased performance, cost-effectiveness, and environmental sustainability in applications such as water treatment. To study the adsorption time, molecularly imprinted polymer membranes were introduced into an aqueous solution containing the 4 template molecules and continuously stirred. Samples were collected at various time intervals (every 10, 20, 30, 40, 50, 60 min, 12 h, and 24 h) and analyzed using HPLC starting from the initial time.

#### 2.3.3. Chromatographic Conditions

Chromatographic separation was achieved by applying a mobile phase containing 0.1% formic acid in water (solvent A) and pure acetonitrile (solvent B). A multi-step gradient elution program was used by using 15% solvent B for the first 3 min at a flow rate of 0.5 mL min^−1^ and at 3.01 min, 60% solvent B was used until 7 min at a flow rate of 0.6 mL min^−1^ and at 10 min, it was changed to 15% solvent B and held at a flow rate of 0.5 mL min^−1^ to 5 min and thereafter changed back 1 mL min^−1^ for equilibration of the system. The total run time for the analysis was 15 min. The UV wavelengths were set according to the absorbance of each compound at 254 and 280 nm. Naproxen, ibuprofen, tenofovir disoproxil and efavirenz were monitored at 254 nm and emtricitabine and diclofenac were monitored at 280 nm. The chromatogram in Figure 2 is a representation of the separation of these compounds studied.

## 3. Results and Discussion

### 3.1. Preparation of MIM and Morphology

Polyethylene Terephthalate (PET) and two variations in polyvinyl alcohol (PVA) with degrees of hydrolysis (DH) of 87–90% and 99% were used. Each polymer was electrospun with different solvents to evaluate their morphologies and suitability for MIP incorporation. For PVA with both DH 87–90% and 99%, 7 g of polymer was used in the electrospinning process. The solvent systems chosen were water and a mixture of ethanol and water (10 mL ethanol, 30 mL water) for both PVA formulations. Additionally, 1% PET was electrospun using trifluoroacetic acid (TFA) and dichloromethane (DCM) solvents in quantities of 2 and 4 mL each. The morphology of the resulting nanofibers plays a crucial role in determining the effectiveness of MIP incorporation. Nanofibers with smoother surfaces, uniform diameters, and higher porosity are desirable for efficient template molecule binding and recognition. The PVA nanofibers produced using water as the solvent exhibited smoother and more uniform structures compared to those spun using the ethanol–water mixture. Differences in fiber diameter, alignment, and surface roughness may influence the MIP incorporation efficiency with varying degrees of hydrolysis. The PET nanofibers spun from TFA in DCM solvents differ in morphology, with TFA producing finer and more uniform fibers with smooth beads.

### 3.2. Characterization of Electrospun MIP Composite Material in Terms of Morphology, Surface Area and Adsorption

#### 3.2.1. Properties of Polymer Solutions

It is important to consider parameters such as conductivity, viscosity, solvent, and temperature when electrospinning polymer solutions with a mixture of molecularly imprinted polymers. These parameters play a significant role in the electrospinning process as they can affect the fiber formation, morphology, and properties of the final product [22]. Conductivity influences the electrostatic interactions during electrospinning, while viscosity determines the solution’s ability to form stable fibers [23]. Temperature can impact the solution’s rheological properties and the electrospinning process. Therefore, by understanding and optimizing these parameters, this can help enhance the electrospinning process and tailor the properties of the polymer–MIP fibers for specific applications.

Table 1 presents various parameters of polymer solutions, including the type of polymer, polymer mass, solvent used, solvent volume, conductivity, and viscosity. When analysing the data, several trends can be observed across different types of polymers and their corresponding MIP and NIP forms. The viscosity measurements were taken at a standardized shear rate of 40 1/s for each sample. Analysis of the data reveals intriguing trends and variations among the different polymer–solvent systems. Firstly, looking at the conductivity values, it is evident that the conductivity of the solutions differs significantly based on the type of polymer and solvent used. For instance, PET solutions in TFA exhibit higher conductivity compared to those in TFA and DCM. This difference can be attributed to the nature of the solvents and their ability to dissociate ions, leading to variations in conductivity levels. TFA has a higher capability of dissociating ions, leading to increased conductivity in the solution as seen in Table 1. On the other hand, the viscosity is lower for the TFA and DCM mixture (3.9) compared to TFA (12.7). This suggests that the addition of DCM has reduced the viscosity of the solution, which is consistent with DCM being a less viscous solvent compared to TFA. DCM is a less polar solvent that does not readily dissociate into ions compared to TFA, which can lead to changes in the ionic mobility and charges present in the solution, ultimately affecting conductivity. Additionally, DCM has different molecular interactions and solvation properties compared to TFA, which can influence the viscosity of the solution. The weak intermolecular forces in DCM allow the molecules to flow more easily past each other, reducing the resistance to flow and thereby lowering the viscosity of the solution. On the other hand, TFA is a polar solvent with stronger intermolecular forces due to the presence of polar functional groups. These stronger intermolecular interactions result in higher viscosity compared to non-polar solvents like DCM, which enhance the solubility of PET. TFA facilitates relatively low viscosity, allowing for smooth flow through the spinneret during electrospinning. This low viscosity, combined with the solvent’s high conductivity, enables the solution to respond effectively to the applied electric field during the electrospinning process. As a result, the electrospun fibers exhibit a uniform and continuous morphology, characterized by smooth surfaces and minimal bead formation. The stronger intermolecular forces in TFA cause the molecules to have more difficulty sliding past each other, leading to increased resistance to flow and higher viscosity in the solution. In contrast, when DCM is introduced into the TFA mixture, the overall conductivity of the solution may decrease due to the lower polarity of DCM compared to TFA. This decrease in conductivity can hinder the electrospinning process by reducing the efficiency of charge injection into the solution, making it more difficult for the fibers to elongate under the electric field. Additionally, DCM significantly influences the viscosity of the solution, potentially leading to a higher overall viscosity when mixed with TFA. The increased viscosity can impede the flow of the solution through the spinneret, resulting in an inconsistent jet and contributing to the formation of an irregular fiber morphology characterized by beads or uneven textures. The lower polarity and different molecular structure of DCM impact the intermolecular interactions within the solution, leading to changes in viscosity [24] as shown in Table 1.

Secondly, the viscosity of the solutions also varied depending on the polymer and solvent combination used. In general, higher viscosity values are observed in PVA DH solutions compared to PET solutions. This difference could be due to the molecular structure of the polymers, where PVA tends to form more entangled chains, leading to increased viscosity. Furthermore, the temperature at which the measurements were taken can influence the conductivity and viscosity of the solutions [25]. Temperature had no influence on these results as everything was measured at room temperature. Additionally, when comparing the MIP and NIP forms of the polymers, it is interesting to note that MIPs generally exhibit higher conductivity and viscosity values compared to their NIP counterparts. This difference can be attributed to the presence of molecular imprints in the polymer matrix, which may alter the interaction of the polymer with the solvent, leading to changes in conductivity and viscosity. This can happen through specific binding interactions between the molecular imprints in the polymer matrix and the target molecule. These interactions can influence the arrangement of polymer chains and solvent molecules, affecting the mobility of ions or charge carriers within the solution, which can impact conductivity. Furthermore, the molecular imprints can create additional cross-linking points within the polymer matrix, resulting in a more rigid structure compared to NIPs. This increased rigidity can affect the flow properties of the solution, leading to higher viscosity values in MIP solutions. Additionally, the presence of molecular imprints can influence the overall structure and porosity of the polymer matrix, which can affect the solvation properties of the solvent within the MIP. This alteration in the solvation environment can impact the flow behaviour and viscosity of the solution [26].

Higher conductivity in polymer solutions improves electrospinnability by enhancing charge transfer and promoting the formation of a stable jet during electrospinning [27], as demonstrated through SEM images of PVA in both degrees using ethanol as a solvent. The fiber appears smoother, more elongated, and exhibits minimal to no bead formation (Figure 3). Conductive solutions tend to promote the formation of finer fibres with reduced bead formation [28], leading to a more uniform fibre morphology. Therefore, polymer solutions with high conductivity, such as those observed in PET solutions in TFA, PET MIP in TFA, and PET NIP in TFA, show electrospun fibres with better quality and uniformity.

On the other hand, varying viscosity levels in polymer solutions can also influence the electrospinning process and the resultant fibre morphology. Higher viscosity solutions exhibit increased resistance to jet elongation and may sometimes lead to the formation of thicker fibres with reduced alignment and uniformity. In contrast, lower viscosity solutions tend to facilitate the formation of finer fibres with improved alignment and reduced defects. PVA solutions have a higher viscosity compared to PET solutions. This difference in viscosity may affect the spinning process during electrospinning, leading to the formation of different morphologies. The higher viscosity of PVA solutions can result in smoother, more elongated fibers with less bead formation compared to fibers produced from PET solutions.

#### 3.2.2. Viscosity vs. Shear Rate Polymer Solution Behaviour

Understanding the shear rate versus viscosity relationship of polymer solutions is important in this study as it helps us understand the behaviour of polymer solutions. The viscosity of a polymer solution varies with shear rate, a phenomenon known as shear-thinning behaviour. This behaviour influences the flow properties of polymer solutions in processing operations such as extrusion, spinning, and coating [29]. By characterizing the rheological behaviour of polymer solutions, optimization formulation, processing parameters, and application properties, leading to the development of advanced materials with enhanced performance and efficiency can be achieved.

Therefore, a viscosity vs. shear rate graph was drawn to see the different behaviours of these solutions in different solvents and when an additive such as MIP and NIP was added to the polymer solution, as shown in Appendix A. Several observations were made that shed light on the rheological behaviour of the polymer solutions studied. All solvents and polymer solutions without molecularly imprinted polymer and non-imprinted polymer additives exhibited Newtonian behaviour, suggesting a linear relationship between shear rate and viscosity. However, a notable exception was observed in the case of PVA 99 DH (polyvinyl alcohol 99% degree of hydrolysis) solution, which displayed shear-thinning behaviour. This can be attributed to the unique structural properties of the PVA 99 DH polymer chains. PVA is known for its excellent film-forming and adhesive properties, as well as its ability to create strong hydrogen bonds with water molecules. In the case of PVA with a high degree of hydrolysis (99%), PVA molecules have a linear structure with hydroxyl (-OH) groups along the backbone [29]. At high shear rates, the polymer chains align in the direction of flow, causing the hydroxyl groups to interact with each other and form temporary hydrogen bonds, which reduces the resistance to flow. This alignment and disruption of hydrogen bonds result in a decrease in viscosity as the shear rate increases. Moreover, in high-hydrolysis PVA solutions, the presence of a high number of hydroxyl groups along the polymer chain enhances the formation of hydrogen bonding between chains, leading to increased intermolecular interactions and higher viscosity at low shear rates. Opposite observations were made by [30]. They noticed that at low levels of viscosity, there is high mobility. This was ascribed to their material, i.e., carboxymethyl cellulose (CMC) blended with PVA, and chemical crosslinking. This is a result of the large number of hydrogen bonds and other weak interactions formed within the hybrid hydrogel network.

The increased hydrophilicity of the PVA 99 DH polymer chains leads to enhanced interactions with the surrounding solvent molecules, particularly water in this case. These interactions result in the formation of hydrogen bonds between the polymer chains and the solvent molecules, creating a network structure within the solution. Under low shear rates, this network structure remains stable, resulting in a higher apparent viscosity as seen in Table 1 at a shear rate of 40, due to the resistance to flow offered by the entangled polymer chains and the hydrogen bonding network. However, as the shear rate increases in the system, the applied force causes the polymer chains to align and orient themselves along the direction of flow. This reorientation of the polymer chains under shear stress disrupts the hydrogen bonds and weakens the physical entanglements within the solutions network, leading to a decrease in viscosity. The shear thinning behaviour observed in the PVA 99 DH solution is a result of this structural rearrangement of the polymer chains in response to shear forces. Furthermore, the high degree of hydrolysis in PVA 99 DH may also contribute to an increased flexibility of the polymer chains, allowing for easier deformation and alignment under shear stress. This higher chain mobility facilitates the reduction in viscosity with an increasing shear rate, as the polymer chains can flow more readily in response to the applied force. A similar shear thinning behaviour of PVA at 99% DH was observed in a study by Reena et al. [31] where they observed lower viscosity of the PVA: resorcinol formaldehyde (RF) gel system that was attributed to a decrease in the semi-crystalline property of PVA due to the presence of salt and the alignment of chain segments of all the samples in the direction of applied stress.

The introduction of MIP and NIP additives into the polymer solutions had no significant impact on their rheological properties. Despite the majority of the MIP and NIP solutions maintaining Newtonian behaviour, the PVA 99 MIP and NIP solutions in both water and water–ethanol mixtures exhibited shear-thinning behaviour. This deviation from the expected rheological response could be linked to the inherent complexity of the polymer–solvent matrix, as well as the presence of specific binding sites created by the molecular imprinting process. The resulting changes in intermolecular interactions and crosslinking densities within the polymer structure may have led to altered flow behaviour under shear stress.

Another factor that could have influenced the shear-thinning behaviour observed in the PVA 99 DH solution is the overall viscosity of the system. It is well established that shear-thinning behaviour is more commonly observed in thick or highly viscous solutions, where the rearrangement of polymer chains under shear stress results in decreased viscosity. In the case of the PVA 99 DH solution, its higher viscosity compared to other polymer solutions may have contributed to the observed shear-thinning behaviour, as the polymer chains experienced more resistance to flow and exhibited a more pronounced response to changes in shear rate. Mongruel and Cloitre [32] and Bounoua et al. [33] observed similar behaviour to the rheological properties described in the study of polymer solutions with various solvents, particularly the shear-thinning behaviour observed in the PVA 99 DH solution, and the research conducted by Mongruel and Cloitre in 1999 [32] on the rheology of polymer solutions. In their study, they investigated the shear-thinning behaviour of polymer solutions and attributed it to the alignment and deformation of polymer chains under shear stress, leading to a decrease in viscosity. They also discussed how the interaction between polymer chains and solvent molecules can affect the rheological properties of the solution, stating that for a given viscosity of suspensions in non-polar solvents, there is much less shear thinning than suspensions in polar solvents. They also noted that shear thinning decreases when the viscosity of the suspending fluid increases, supporting the observations made in the study of polymer solutions with various solvents.

### 3.3. Effects of Voltage

The optimization of applied voltage in the electrospinning process plays an important role in controlling the formation and properties of nanofibers. By carefully adjusting the voltage, the morphology, diameter, alignment, and uniformity of the nanofibers produced can be influenced [34]. In this experiment, the behaviour of polyvinyl alcohol (PVA) and polyethylene terephthalate (PET) nanofibers was studied at a constant flow rate, and tip-to-collector distance to investigate the impact of applied voltage on the electrospinning process. For voltage effects, both polymer solution images represent each voltage stage for the purpose of this discussion.

Experimental tests were conducted with a range of applied voltages spanning from 10 kV to 18 kV. The outcomes of these experiments are depicted in Figure 3a–c. The data revealed that the average diameter of fibers increased slowly as the voltage increased. At the voltage of 10 kV for a PET/TFA solution, a large number of beads with thin fibers were formed (Figure 3a,d). These fibers had a diameter of ~45 nm. This occurs because as the electric field strength increased, the repulsive forces between the charges on the droplet surface also increased, which leads to a greater stretching force on the droplet, causing the fibers to become thinner. However, when the voltage was increased to 15 kV, there was a significant proportion of fibers with diameters below 150 nm, indicating the production of thicker fibers with less beads (Figure 3b,e). The electric field strength was directly proportional to the voltage applied. When increasing the voltage, the electric field strength increased, which led to thinner fibers.

At 18 kV of both PET and PVA polymers, the distribution of fiber diameters broadened, leading to the formation of thicker fibers. The observed results from the frequency distribution data and PET/TFA showed varying patterns across different fiber diameter ranges exhibiting a more even distribution across the 75–200 nm range (Figure 3c). Similar results were attained for PVA under the same experimental conditions (Appendix A). A study by Faizal et al. (2020) [35] showed similar behaviour when they varied the voltage between 12 kV and 18 kV at 0.2 mL/h. The fiber diameter was 36.90 ± 7.00 µm and the fibers had various size diameters of both thin and thick fibers with the use of a higher voltage as used in this study. The difference in polymers solutions led to distinct fiber diameter profiles, possibly due to variations in polymer chain length and cross-linking density because of the differences in the length of the polymer chains and how tightly they are connected in each polymer. For instance, PET/TFA with a low molecular weight ranging between 8000 and 31,000 kDA showed a distribution skewed towards the larger fiber diameter ranges, with the 150–175 nm range having the highest frequency at a higher voltage. Specifically, it was observed that a substantial proportion of thin fibers with diameters below 150 nm were evident when the applied voltage surpassed 15 kV. An increase in the applied voltage, corresponding to an increase in the electric field intensity, led to a heightened electrostatic repulsive force acting on the fluid jet, thereby promoting the formation of thicker smooth fibers [36]. Conversely, the expedited ejection of the jet from the Taylor cone results in a swifter removal of the solution from the capillary tip, consequently leading to an enlargement of the fiber diameter. Similarly, the use of PET with TFA as the solvent resulted in larger fiber diameters compared to the PVA materials, which may be attributed to differences in molecular weight, structure and intermolecular interactions. In the case of PET, its molecular structure consists of repeating units of terephthalic acid and ethylene glycol, resulting in a linear polymer chain. This linear structure allows for stronger intermolecular interactions, such as Van der Waals forces, hydrogen bonding, and dipole–dipole interactions between adjacent polymer chains. These interactions give PET its high tensile strength and resistance to deformation [37]. On the other hand, PVA has a larger but more flexible molecular weight ranging between 30,000 and 70,000, and structure due to the presence of hydroxyl groups on its backbone [38]. This flexible structure results in weaker intermolecular interactions compared to PET, leading to the formation of smaller fiber diameters as the polymer chains are less constrained and can orient themselves more easily during fiber formation, leading to increased viscosity and lower conductivity, as seen in the table in Section 2.2, which then affects the electrostatic forces acting on the solution. A more conductive solution will allow for better charge transfer between the spinneret and the collector, resulting in a more stable electrospinning process and finer, more uniform fibers. Therefore, the differences in molecular weight and structure and intermolecular interactions between PET and PVA play a significant role in the voltage applied.

### 3.4. Effects of Flowrate

The choice of flow rate is important in the electrospinning process as it directly impacts the morphology, diameter, and alignment of the electrospun fibers. An optimal flow rate ensures a consistent, uniform spinning process, controlling the amount of polymer solution ejected from the spinneret and determining the rate at which the fibers are formed [39]. Too high of a flow rate can lead to the formation of thicker fibers with potential bead defects, while a flow rate that is too low may result in unstable jet formation and irregular fiber deposition. The flowrate was investigated using a flowrate ranging between 0.2 and 0.4 mL/h, while all other parameters were held constant as seen in Figure 4a–c. At a flowrate of 0.2 mL/h, finer and more uniform fibers with a few beads were observed due to better stretching and alignment of the polymer chains, giving enough time for solvent evaporation to occur (Figure 4a). The morphology of PVA nanofibers produced a flowrate of 0.3 mL/h as shown in Figure 4b; a slightly higher polymer flow introduces more material, which leads to moderately thicker fibers compared to the lower flow rate. The applied voltage continued to facilitate the stretching and alignment of the polymer chains, contributing to the formation of well-defined fibers with a higher aspect ratio.

However, the moderate increase in flow rate resulted in some irregularities along the fibers with fewer thin fibers formed. The results showed that the fibers were relatively thick, with an average diameter of approximately 200 nm, and exhibited a few observed beads as shown in Figure 4d. This morphology is not considered an issue, as the fibers are still suitable for use in various applications. The thick fibers and minimal bead formation are expected to influence the adsorption properties of the nanofibers. In contrast, the morphology at a flow rate of 0.4 mL/h led to entanglements and irregularities in the resulting fibers, impacting the uniformity and quality of the fibers produced compared to the flowrate of 0.3 mL/h. The rapid polymer flowrate overwhelms the electrostatic forces exerted during spinning and the polymer chains do not have sufficient time to align and orient properly before solidification, resulting in the observed entanglements, kinks, and irregularities in the fibers as shown in Figure 4c. The thicker fibers produced at the flowrate 0.3 mL/h may have improved mechanical properties, such as tensile strength and Young’s modulus, compared to the thinner fibers produced at the lower flowrate. This is because the thicker fibers have a higher density of polymer chains, which can improve their mechanical properties; hence, this flowrate was chosen for future experiments. Other researchers have also investigated the effect of flowrate on the morphology of PVA nanofibers. For example, a study by Singh et al. (2020) [40] found that increasing the flowrate from 0.4 mL/h to 0.6 mL/h produced similar results with a significant increase in fiber diameter and bead formation. The authors attributed this to the increased polymer concentration at the tip of the needle, which led to the formation of larger droplets that solidified into thicker fibers.

Another study by Baykara and Taylan (2021) [41] found that increasing the flowrate from 0.2 mL/h to 0.4 mL/h resulted in a decrease in fiber diameter and an increase in bead formation. The authors attributed this to the increased polymer solution velocity at the tip of the needle, which led to the formation of smaller droplets that solidified into thinner fibers. A higher flowrate resulted in an increased amount of polymer solution being delivered per unit time, which influenced the jet stability, stretching, and solidification processes during fiber formation.

### 3.5. Needle Effect

The choice of needle gauge plays a crucial role in the electrospinning process, as it can significantly impact the morphology and properties of the resulting fibers [42]. The effect of needle gauge on fiber diameter is investigated in this study, while all other parameters were held constant. In this study, three needles, viz. 23 G with an inner diameter of 0.337 mm, 18 G with an inner diameter of 0.838 mm and 11 G with an inner diameter of 2.388 mm, were used to investigate the effect of needle size. The 23 G and 18 G needles produced fibers with smaller diameters due to the enhanced electric field intensity around the needle tip (Figure 5a,b,d). The enhanced electric field around the needle is caused by the sharpness and small radius of curvature of the needle tip, which results in a higher concentration of electric field lines. Xie and Zeng (2012) [43] made similar observations when they studied the multi-needle electric field effect to achieve thin fiber diameters. They concluded that the strength of the electric field of the jets on the sides was too strong to form a stable Taylor cone.

This contrasts with the 11 G needle, which produced fibers with larger diameters. The electric field intensity around the needle tip is lower for the 11 G needle with a larger inner diameter compared to the smaller 23 G and 18 G needles. This is because the larger inner diameter of the 11 G needle results in a less sharp and less curved tip, leading to a more diffuse distribution of electric field lines and a lower intensity of the electric field in that region. The lower electric field intensity around the tip resulted in a polymer solution being drawn-out, leading to thicker uniform fibers being deposited onto the collector (Figure 5c). Additionally, the larger inner diameter needle was more effective in spinning polymer solutions with a mixture of the MIP and NIP.

### 3.6. Tip-to-Collector Effect

The tip-to-collector distance (TCD) also plays a crucial role in the electrospinning process as it determines the flight time of the jet, the stretching of the polymer solution, and the deposition pattern on the collector [44,45]. The shorter TCD led to a formation of beads due to the charged jet not having enough time to stretch and solidify before reaching the collector resulting (Figure 6a). In this case, the TCD was kept at 8 cm with the resulting fibers having diameters below 100 nm. The resulting beaded fibres are formed as the jet did not have sufficient time for stretching because of the short flight time. This meant that the solvent did not have time to evaporate, resulting in the formation of beads. However, the increase in the TCD created more space for elongation and stretching, thus leading to smooth fibers without beads. When the TCD is 10 cm, there are smooth fibers with less beads (Figure 6b). The resulting fibers had an average diameter ranging from 300 to 350 nm. In the case of 15 cm TCD, the fiber diameter was about 350 nm, as seen in Figure 6c. There was enough time for solvent evaporation from the charged jet, resulting in smooth fibers. In addition, the distance allowed the stretching of the jet. Purwar et al. (2016) [46] reported that the shorter TCD lead to the formation of beads without a fibrous structure being observed. The authors observed smooth fibers when the distance was increased from 3 cm to 8 cm; they attributed such a behaviour to the charged jet being stretched enough prior to the deposition to the collector. They indicated that a shorter TCD led to a shorter flight distance for the jet. This resulted in uneven stretching and incomplete solvent drying of the jet before reaching the collector, thus resulting in beaded fibers.

### 3.7. Effects of Polymer Solvent

The importance of studying the effects of the solvent used for polymer solutions, such as PVA and PET, lies in its significant impact on the properties, processing techniques, and environmental sustainability of the resulting materials. The choice of solvent influences the dissolution process of the polymer, molecular structure, and material properties, leading to variations in mechanical, and thermal characteristics [47]. Additionally, solvent selection affects processing parameters, such as fiber diameter and morphology, as well as environmental considerations like solvent recycling and waste generation. Understanding solvent–polymer interactions is essential for optimizing material performance, processing methods, and promoting eco-friendly manufacturing practices in the polymer industry [48]. In this section of the study, the solvents used to dissolve PET and PVA will be examined.

Therefore, this study investigated the impact of various solvents on PVA and PET polymers. The solvents used include water and a water–ethanol mixture (3:1 ratio) for PVA, and trifluoroacetic acid (TFA) and a TFA-dichloromethane (DCM) mixture (3:1 ratio) for PET. In the case of PVA in water, it was noted that water is a good solvent with good compatibility and hydrogen bonding capabilities with PVA, which helps in the effective dissolution of the polymer. Water is known to have lower surface tension compared to organic solvents, which lead to improved fiber morphology and reduced bead formation during the electrospinning process [49]. The fibers exhibited a smooth surface with minimal bead formation, showcasing fiber formation supported by hydrogen bonding as illustrated in Figure 7a.

The addition of ethanol to a PVA solution during electrospinning has multiple effects on the process and the resulting fiber formation. Ethanol acts as a plasticizer, reducing intermolecular forces and increasing polymer chain mobility, leading to decreased solution viscosity and improved spinnability [50]. The lowered surface tension caused by ethanol helped reduce bead formation and enhance the uniformity of the fibers. The presence of ethanol also influences drying kinetics, promoting faster solvent evaporation (Figure 7b). Water is a commonly used solvent for PVA due to its high solubility and non-toxic nature. The presence of ethanol in water can alter the solvent properties, such as viscosity and surface tension, affecting the solution’s spinnability. Ethanol can enhance the processability of the PVA solution by improving polymer–solvent interactions, resulting in uniform fiber production. Furthermore, the use of water-based solvents like water and ethanol in water for PVA samples contributes to the eco-friendliness of the spinning process, as water is a renewable and environmentally friendly solvent compared to organic solvents.

In the case of PET in TFA, TFA is a very strong solvent. The use of it promoted chain entanglement and charge repulsion within this polymer, producing beaded thin fibers (Figure 7c). As a solvent, TFA effectively dissolves PET, resulting in a homogeneous solution with a specific viscosity suitable for electrospinning. The controlled viscosity provided by TFA plays an important role in the formation of fibers. The low viscosity of the solution affected jet stability, stretching behaviour, and the ability of the fibers to solidify and maintain their shape during electrospinning. The low viscosity does not provide enough resistance for the jet’s elongation, leading to the formation of droplets that solidify into beads rather than fiber. Furthermore, the low conductivity of this solvent influenced the formation of the beaded structure. The low conductivity led to poor ionization of the solution, resulting in a weak electrostatic force that is needed to draw and stretch the polymer solution into fibers. In this solution, the conductivity and electrospinnability of the solution were enhanced, as mentioned in Section 2.2 (Table 1). The addition of DCM in this mixture affected the morphology of the fiber. They were highly beaded with very thin fibers that broke under imaging (Figure 7d). The addition of DCM further reduced the viscosity and conductivity of the solution that caused the morphology of the polymer to increase bead formation.

TFA is a strong acid with high solvating power, which is commonly used for PET due to its ability to dissolve the polymer effectively [51]. However, TFA is challenging to handle due to its corrosive nature and toxicity. The addition of DCM in TFA modifies the solvent properties, such as volatility and evaporation rate, influencing the spinnability of the PET solution. A study by Mahalingam et al. (2015) [52] investigated the spinnability and solubility of PET with different solvents. They pointed out that PET is only spinnable when using TFA and/or DCM. The beaded fibers were formed with the use of TFA alone, which corresponds with the observations in this study. However, the addition of DCM produced smooth nanofibers, which is in contrast with this study’s observation of increased beads and slightly elongated thin fibres.

However, the use of TFA-based solvents for PET samples poses environmental considerations due to the toxicity and potential hazards associated with TFA. Proper waste management and handling protocols are essential to minimize environmental impact when using TFA-based solvents in polymer processing. Since the intended application is to treat wastewater sites, the choice of solvents for polymer solutions becomes critical in terms of environmental consideration and eco-friendliness. In this research, the production of the electrospun polymer membrane must utilize environmentally sustainable methods to prevent the generation of detrimental by-products that could exacerbate the existing pressure on water resources. Therefore, PVA was used as the material of choice because it can be dissolved in water and other eco-friendly solvents. Conversely, TFA-based solvents like TFA and DCM in TFA for PET samples may raise concerns regarding their toxicity and potential environmental harm if not properly managed. Therefore, due to the environmental considerations and eco-friendliness associated with water-based solvents, further studies were conducted exclusively using polyvinyl alcohol (PVA) to form molecularly imprinted membranes (MIM) and nonimprinted membranes (NIM). The selection of PVA as the base polymer for these studies aligns with the goal of promoting sustainable practices in polymer processing and applications. Additionally, the use of PVA-based polymer for the MIM and NIM applications demonstrates a commitment to utilizing environmentally friendly materials in the development of innovative technologies. By focusing on PVA as the polymer of choice for these applications, the research contributes to the advancement of eco-friendly solutions in the field of polymer science.

### 3.8. Effects of PVA Polymer Choice by Degree of Hydrolysis

Determining the optimal degree of hydrolysis for PVA was essential for achieving desirable material properties and processability. When comparing PVA with degrees of hydrolysis of 87–90% and 99%, considering their respective advantages and disadvantages is crucial. Analysing the data presented in Table 2 and Section 2.2 (Table 1) for PVA solutions of different degrees of hydrolysis (DH) dissolved in water, PVA with a DH of 87–90% displayed better conductivity, spinnability, and viscosity when compared to PVA with a DH of 99%, thus resulting in the formation of smooth fibers. The PVA solution with a DH of 87–90% exhibited lower conductivity (1054 µS/cm) compared to the 99% DH solution (839 µS/cm), which is beneficial for jet stretching during electrospinning and ensuring uniform thinner fiber deposition.

Furthermore, the PVA 87–90% solution demonstrates higher spinnability (165.5 mPa·s viscosity) compared to the 99% DH solution (2058.8 mPa·s viscosity), indicating better processability and fiber alignment. The lower viscosity of the 87–90% DH solution facilitates smoother jetting of the polymer solution, leading to the formation of well-defined and stable polymer mats. PVA with a lower degree of hydrolysis (87–90%) typically exhibits higher molecular weight chains and increased chain entanglements, and thus high viscosity [53].

On the other hand, PVA with a higher degree of hydrolysis (99%) offers better water solubility and compatibility due to a higher number of hydroxyl groups. However, the increased hydrophilicity of PVA 99% affects its suitability for certain applications. For instance, filtration membranes must exhibit certain mechanical performance, and high hydrophilicity may result in a reduction in mechanical strength and stability, and thus limit its application in wastewater treatment. In terms of electrospinning to form polymer mats, looking at the polymer properties of PVA 87–90% mentioned in Section 2.3 and comparing them to those in Table 2, these results show that PVA with a lower degree of hydrolysis (87–90%) is preferred due to its superior mechanical properties and enhanced electrospinnability compared to PVA 99%. Research by Park et al. (2010) [54] supported this finding, demonstrating that PVA 87–90% exhibited slightly higher mechanical strength and improved fiber formation during electrospinning than those below 87%. They further experienced the difficulty and poor electrospinnability of the PVA 99% as was experienced in this study, stating that when DH = 99.9%, because of higher surface tension, it has a tendency to undergo gelation through strong hydrogen bonding. Mostly beaded fibers were formed with DH = 99%; similar results were reported by Zhang et al. (2005) [55].

Considering the advantages of PVA 87–90% in terms of mechanical strength and electrospinnability, a polymer mat of PVA 87–90% was selected to incorporate MIPs and NIPs in the formation of PVA-MIP-based composites for the adsorption of select pharmaceuticals in water treatment applications. This choice was made to capitalize on the superior properties of PVA 87–90% for achieving improved performance and stability in pharmaceutical adsorption processes.

### 3.9. Interpretation of the Results from the Adsorption and Application Studies

#### 3.9.1. Material Swelling Studies

The results obtained from the centrifugation experiment conducted on the PVA 87–90 DH electrospun MIM and NIM revealed an interesting trend in the weight of the materials over different time intervals. For the MIM, the weight decreased as the centrifugation time increased, starting at 2.864 g at 5 min and decreasing to 1.6483 g at 60 min before completely dissolving at 2 h. On the other hand, the NIM also showed a decrease in weight over time, starting at 1.404 g at 5 min and diminishing to 0.136 g at 60 min before complete dissolution at 2 h. This trend in weight loss can be explained based on the chemistry of PVA. PVA is a water-soluble polymer that swells in water and can dissolve when exposed to prolonged contact with water. The decrease in weight of both MIM and NIM membranes over time can be attributed to the continuous hydration and swelling of PVA molecules in contact with water during centrifugation. This hydration causes the polymer chains to separate and dissolve, leading to the observed weight loss. Additionally, the presence of molecular imprints in the MIM could potentially affect the dissolution behaviour compared to the NIM, as the specific interactions between the imprinted sites and target molecules influence the swelling and dissolution properties of the membrane. Further investigation into the molecular imprinting process and the effects of specific interactions on the dissolution behaviour could provide valuable insights for the development and optimization of PVA-based membranes for various applications. Figure 8 shows the swelling study graph.

#### 3.9.2. TGA Spectra of Polymer Materials

Understanding the thermal stability of polymer materials is important for various applications. The choice of polymer material can significantly impact the adsorption efficiency, selectivity, and stability, making thermal stability a critical consideration. The thermal stability of a polymer material is important in adsorption applications because it determines the material’s ability to withstand temperature fluctuations and maintain its adsorption properties over an extended period. Polymer material with a high thermal stability can ensure consistent performance and minimize the risk of thermal degradation.

A comparative discussion of the polymer materials formed in the TGA graphs in Figure 9a–c reveals distinct weight loss or degradation points in each material. While all the polymer materials exhibit thermal degradation, the temperature points at which this occurs differ significantly between them. For instance, PVA 99 ETOH: water, a polyvinyl alcohol-based polymer, showed a relatively narrow temperature range for weight loss, with significant degradation occurring between 100 °C and 150 °C due to the loss of water and ethanol. The most pronounced weight loss was observed around 150–270 °C; as the temperature increases towards 270 °C and beyond, the weight loss may indicate the breakdown of the polymer backbone and further degradation of the PVA molecules as seen in Figure 9c. The complete degradation observed at around 460 °C suggests the decomposition of PVA into smaller molecular fragments, indicating the end of the thermal stability of the polymer. Overall, the weight loss at these specific temperature points can be linked to the evaporation of moisture or solvent molecules, as well as the thermal degradation of the PVA polymer chains.

In contrast, PET TFA, a polyethylene terephthalate-based polymer, displays a more extensive temperature range for weight loss, with significant degradation occurring between 230 °C and 405 °C. The initial weight loss of around 150 °C is likely due to the dehydration of PET, which is a common phenomenon observed in many polymer systems. The second weight loss peak around 405 °C could be related to the thermal degradation of PET’s molecular structure, possibly involving the breakdown of its ester linkages or the formation of volatile compounds. The complete degradation around 460 °C is due to the destruction of the PET polymer chain, resulting in a significant loss of mass. In the case of PET-TFA, weight loss at these temperatures could be influenced by the presence of TFA, which is known for its strong acid properties. PET TFA/DCM, a blend of polyethylene terephthalate and dichloromethane, exhibits a similar temperature range for weight loss as PET TFA, with significant degradation occurring between the same temperature ranges as seen in Figure 9. However, the most pronounced weight loss is observed around 230 °C and 400 °C, which indicates different thermal degradation mechanisms compared to PET TFA. The presence of dichloromethane influences the thermal behaviour of this material at this temperature range. One possible reason for this is that DCM is a solvent that can affect the molecular structure and interactions within the polymer as mentioned in previous sections. When DCM is present in the polymer, it can disrupt the hydrogen bonding and π-π stacking interactions between the polymer chains, leading to a more open and flexible structure. This increased flexibility can make the polymer more susceptible to thermal degradation, as the polymer chains can move more easily and break apart more easily. The MIP and NIP possess complex 3-dimensional structures formed as a polymer network by pyridine, which makes them thermally stable polymer materials. The presence of the heterocyclic aromatic pyridine ring is a significant contributor to their stability, as it provides a rigid and planar structure that is resistant to thermal degradation. When the MIP and NIP are mixed with polymer solutions, such as PVA or PET, they can shield the polymers from thermal degradation. This is evident in the TGA diagrams in Figure 9a,b, which show a delayed thermal degradation of the PVA-MIP and PET-MIP, as well as PVA-NIP and PET-NIP. There is about 21–51% delayed degradation when the MIP is added to the PVA, and about a 4–8% delayed shift to PET-MIP materials. The MIP and NIP act as a thermal barrier, preventing the polymer chains from undergoing thermal degradation. This shielding effect is attributed to the 3-dimensional structure of the MIP and NIP, which provides a protective environment for the polymer chains. The MIP exhibits a unique temperature range for weight loss with significant degradation occurring between 300 °C and 460 °C. The most pronounced weight loss is observed around 100 °C and 280 °C, suggesting a more gradual thermal degradation process of water and ethanol evaporation as well as the initial collapse of the MIP material as seen in the previous Section 2.3 the collapse of the MIP backbone. This broader temperature range may indicate a more complex and variable thermal behaviour for this material compared to PVA 99 ETOH: water as seen in Figure 9a.

In comparison to the other materials, PVA 99 ETOH: water exhibits a relatively low temperature range for weight loss, indicating a more stable thermal behaviour. PET TFA and PET TFA/DCM exhibit broader temperature ranges for weight loss, suggesting more complex and variable thermal behaviours. MIP exhibits an intermediate temperature range for weight loss, indicating more complex thermal behaviour compared to PVA 99 ETOH: water but less complex than PET TFA and PET TFA/DCM. The NIP polymer mixture materials exhibit unique thermal behaviour. Specifically, the NIP polymer mixture with PVA 99 ETOH: water exhibits a distinct weight loss pattern, with significant degradation occurring between 100 °C and 270 °C. The most pronounced weight loss is observed around 270 °C, suggesting a thermal degradation event.

In comparison to the other polymer materials, the NIP–polymer mixture with PVA 99 ETOH: water exhibits a relatively high temperature range for weight loss, indicating more complex thermal behaviour. This may be due to the presence of the NIP–polymer mixture, which can influence the thermal stability of the PVA 99 ETOH: water. The NIP–polymer mixture with PET TFA exhibits a similar temperature range for weight loss to PET TFA alone, but with a more pronounced weight loss around 100 °C and 260 °C. This may indicate a synergistic effect between the NIP–polymer mixture and PET TFA as seen in Figure 9b. A study by Miranda et al. (2017) [56] found that when LA (lactic acid) was blended with PET powder, the weight loss began at 130 °C, likely due to the evaporation of LA from the PET surface. As the processing steps progressed from pellets to preforms and bottles for the two-stage process, the weight loss at 250 °C decreased. This decrease in weight loss was used as an indicator of the concentration of unbound LA, which was lower than the nominal loading. The two-stage preforms showed lower weight loss than pellets, consistent with previous studies on end groups and extraction.

In contrast, the single-stage process preforms showed a higher weight loss at 250 °C, with a total loss of 0.33% at 120 °C. This suggests that LA in the PET matrix migrated during extraction in the single-stage process. This finding is supported by end-group analysis, which showed greater reaction between end groups to form LA-capped PET for the two-stage process.

In contrast, the NIP–polymer mixture with MIP exhibits a relatively low temperature range for weight loss, indicating a more stable thermal behaviour. The most pronounced weight loss is observed around 80 °C, suggesting a thermal degradation event. The NIP–polymer mixture with PET TFA/DCM exhibits a complex temperature range for weight loss, with significant degradation occurring between 200 °C and 400 °C. The most pronounced weight loss is observed around 400 °C and 460 °C, suggesting multiple thermal degradation events. A study by Yang et al. (2021) [57] revealed that PVA undergoes significant thermal degradation between 216 °C and 320 °C, resulting in substantial weight loss due to the breakdown of its polymer backbone. Notably, the highest rate of weight loss occurred at 278 °C. In contrast, a lower temperature of approximately 130.3 °C led to a minor weight loss of around 5% due to dehydration. When subjected to high temperatures, PVA decomposes completely, leaving no residue behind. This decomposition is attributed to oxidative processes involving organic substances in an air-containing atmosphere. Factors such as molecular weight, molecular structure, and degree of crystallinity can influence the degradation of PVA, while its state (molten or solid) also impacts its thermal degradation, resulting in distinct TG profiles.

In conclusion, the comparative discussion of the polymer materials reveals distinct weight loss or degradation points in each material. The temperature ranges and points of weight loss differ significantly between the materials, indicating varying levels of thermal stability and complexity. While PVA 99 ETOH: water exhibits relatively stable thermal behaviour, PET TFA and PET TFA/DCM exhibit more complex thermal behaviours. MIP exhibits an intermediate level of thermal complexity. Understanding these differences can inform the selection of appropriate polymer materials for specific applications where thermal stability is critical.

#### 3.9.3. FTIR Spectra of Polymer Materials

The FTIR characterization of PVA and PET in different solvents, including water, ethanol in water, TFA, and DCM in TFA, as well as solutions of polymers with MIP and NIP additives, provides a comprehensive understanding of the chemical interactions and functional group compositions present in the synthesized nanofiber structures. Analysing the molecular bonding and characteristic peaks in the FTIR spectra allows for the identification of specific functional groups such as O-H, C-H, CH_2_, C-O, and C-C, revealing the intricate molecular arrangements and intermolecular connections within the polymer matrices. This detailed FTIR analysis offers valuable insights into the polymer–solvent interactions and polymer–polymer bonding, shedding light on the complex chemistry involved in the formation of nanofibers with tailored properties and applications. The MIP and NIP functional groups present have been discussed alone in Section 2.1; therefore, in this section, they are only discussed in comparison to the other polymer solution.

Figure 10a–c illustrate the FT-IR results, showing the presence of various functional groups such as O-H, C-H, CH_2_, C-O, and C-C in the synthesized nanofibers. Specifically, the stretching vibration of the O-H group was observed in the range of 3000–3800 cm^−1^, with results showing a peak at 3275 cm^−1^. The O-H symmetry stretch vibration appeared at a wavenumber of 3336 cm^−1^. Furthermore, the vibration of the C-H group was detected at 2910 cm^−1^, while the asymmetrical vibration of the C-H group occurred at 2900 cm^−1^. The CH_2_ group vibration was found at 1430 cm^−1^, with experimental results showing a peak at 1420 cm^−1^. The C-H group vibrations were observed at 1330 and 1377 cm^−1^, with a distinct peak at 1331 cm^−1^. The absorption band at 1078 cm^−1^ indicated the presence of the C-O stretch in the nanofibers. Additionally, the vibration at 849 cm^−1^ denoted a C-C PVA stretch, with experimental results showing a peak at 857 cm^−1^. These stretches and vibrations were consistent with those found by various researchers (Ullah et al., 2020; Zhao et al., 2017) [58,59].

The FTIR spectra of PET in TFA and PET in TFA/DCM solutions in Figure 10c reveal characteristic absorption bands corresponding to specific functional groups present in the polymer films. In the PET/TFA system, notable absorption peaks were observed at approximately 2959 cm^−1^, indicative of C-H aliphatic groups. Another prominent peak at around 1718 cm^−1^ was attributed to the -C-O bond within the PET structure prominent absorption bands, which include the stretching vibration of the carbonyl group (C=O) in PET at around 1715 cm^−1^. This peak indicates the presence of ester groups in the polymer chain, which are essential structural components of PET due to the ester linkages between terephthalate and ethylene glycol units in the polymer chain, while peaks at approximately 1540 cm^−1^ and 1457 cm^−1^ suggested the presence of benzene rings disubstituted in the PET polymer. Additionally, peaks at approximately 1281 cm^−1^, 1203 cm^−1^, and 1121 cm^−1^ corresponded to the -C-C-O asymmetric stretch, the -O-C-C asymmetric stretch, and aromatic ring vibrations, respectively. A peak at 1073 cm^−1^ was associated with a benzene para-substituted group in the PET structure, while another peak at 843 cm^−1^ indicated C-H vibrations within the aromatic structure peaks around 1380–1450 cm^−1^, corresponding to the bending vibrations of the methyl (-CH_3_) and methylene (-CH_2_-) groups in PET. These peaks provide further evidence of the molecular structure of PET and can be used to confirm the presence of these alkyl groups within the polyester chain. Furthermore, a peak at 724 cm^−1^ was observed in the FTIR spectrum of PET in TFA, although the specific functional group responsible for this vibration was not explicitly identified. Similar stretching and vibrations were found by Nojavan et al. (2024) [60] while studying PET in combination with carbon nanotubes.

#### 3.9.4. Adsorption Time of the Electrospun PVA-MIP and PVA-NIP Composites

The adsorption behaviour of various drugs (EMI, TENO, NAP, DICLO, IBU, EFV) on both molecularly imprinted membrane (MIM) and non-imprinted membrane (NIM) materials over different time intervals (5, 10, 20, 40, 60 min) was investigated. The data reveal the amount of each drug adsorbed onto the materials at each time point, providing insights into the adsorption kinetics of the drugs. Figure 11 portrays the pharmaceutical adsorption process using an electrospun-based PVA-MIP material over different time intervals. As expected, the results indicate a gradual increase in pharmaceutical adsorption as the time of contact between the material and the solution increases. This is consistent with the typical behaviour of adsorption processes where the adsorption capacity of the material increases over time, eventually reaching a plateau where equilibrium is achieved. Starting with EMI, the graph shows that EMI exhibits higher adsorption onto MIM compared to NIM for all time intervals. Similar results were reported in a PET-MIP-based material to adsorbent rhodamine blue dye in river water [61]. This suggests that the MIM material may have specific recognition sites for EMI, enhancing its adsorption capacity. The trend is consistent across all drugs, indicating the effectiveness of molecular imprinting in enhancing selectivity and adsorption efficiency.

For TENO, NAP, DICLO, IBU, and EFV, similar trends are observed, with higher adsorption onto MIM compared to NIM. This consistent pattern further validates the benefits of molecular imprinting in enhancing adsorption performance. The data also reveals that the adsorption levels generally increase with time, indicating that a longer contact time results in higher drug uptake by both NIM and MIM. However, it is important to note that the adsorption capacity of the materials may be limited by factors such as stability and durability. In this context, the data show that the PVA-MIP material dissolved completely after 60 min, highlighting the importance of material selection and stability in adsorption studies. The dissolution of the PVAMIP material limits its practical utility as an adsorbent in water treatment applications, necessitating the need for material improvements to enhance stability and performance. One possible reason for this dissolution could be the sensitivity of PVA to prolonged exposure to the solution conditions. To overcome this issue of dissolution, crosslinking the PVA material with other polymers that are hydrophobic might help to maintain the stability of the material. A study by Zhao et al. (2017) [59] showed that crossing PVA with sericin improved the adsorption of organic dyes in water as well as heavy metals.

## 4. Conclusions

The synthesis and characterization of electrospun molecularly imprinted polymer composite membranes, incorporating both polyvinyl alcohol and polyethylene terephthalate, present a promising approach for the remediation of pharmaceutical contaminants in wastewater. This study highlighted the potential of electrospun fiber technology to enhance selective adsorption capabilities due to the tailored design of MIPs, which provide specialized binding sites for target pollutants.

The comparative analysis of PVA and PET in relation to their electrospinning behaviour, fiber morphology, and rheological properties highlights the significance of molecular interactions and hydrolysis degrees in determining the final properties of the nanofibers. The uniformity and smoothness of PVA fibers, especially those with a degree of hydrolysis of 87–90%, were better compared to those of PET, leading to better spinnability and fiber formation. Additionally, key parameters such as solvent choice, electrospinning voltage, flow rate, and needle gauge were shown to critically impact fiber morphology, affecting adsorption performance. A high voltage of 15 KV, slower flowrate of 0.3 mL/min, and 11 G needle were found to be the optimum variables in forming thick and uniform fibres for effective adsorption.

The study’s results also show that MIP composite membranes exhibit enhanced adsorption capacities for pharmaceuticals, validating the efficacy of molecular imprinting in creating materials with specific target recognition. The improved selectivity and adsorption kinetics of the MIMs for various pharmaceutical compounds, including antiretroviral drugs and non-steroidal anti-inflammatory drugs, confirm their viability as advanced filtration media.

However, challenges regarding the dissolution and stability of PVA under continuous water exposure highlighted the necessity for further optimization, such as potential crosslinking with hydrophobic polymers to enhance durability. The use of TGA and FTIR analyses provided insightful data on thermal stability and chemical interactions that underline the robustness of the synthesized materials.

## Figures and Tables

**Figure 1 polymers-16-03297-f001:**
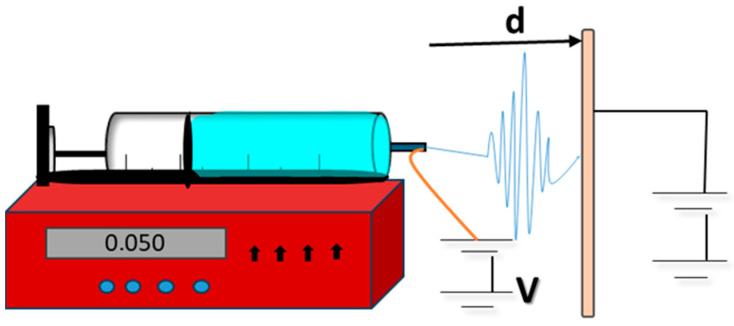
A basic setup schematic for an electrospinning experiment with a horizontal electrode arrangement. d (distance) and V (voltage).

**Figure 2 polymers-16-03297-f002:**
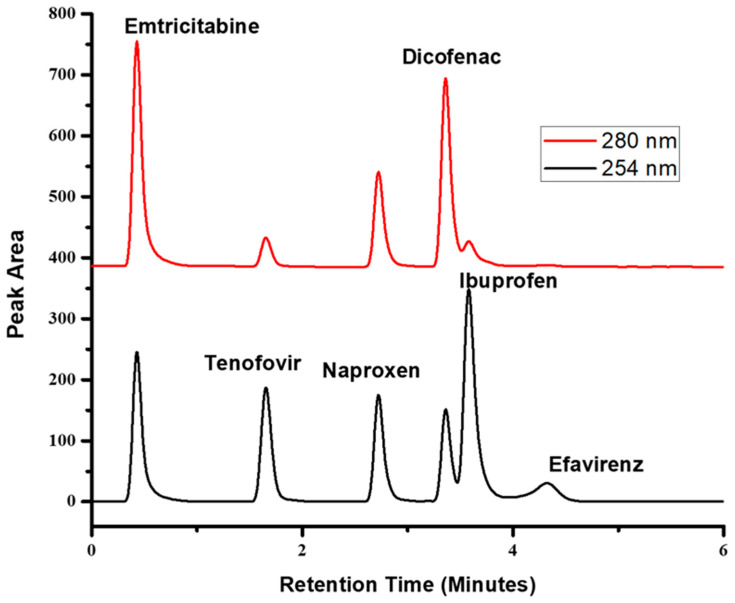
Retention time and peak area chromatogram.

**Figure 3 polymers-16-03297-f003:**
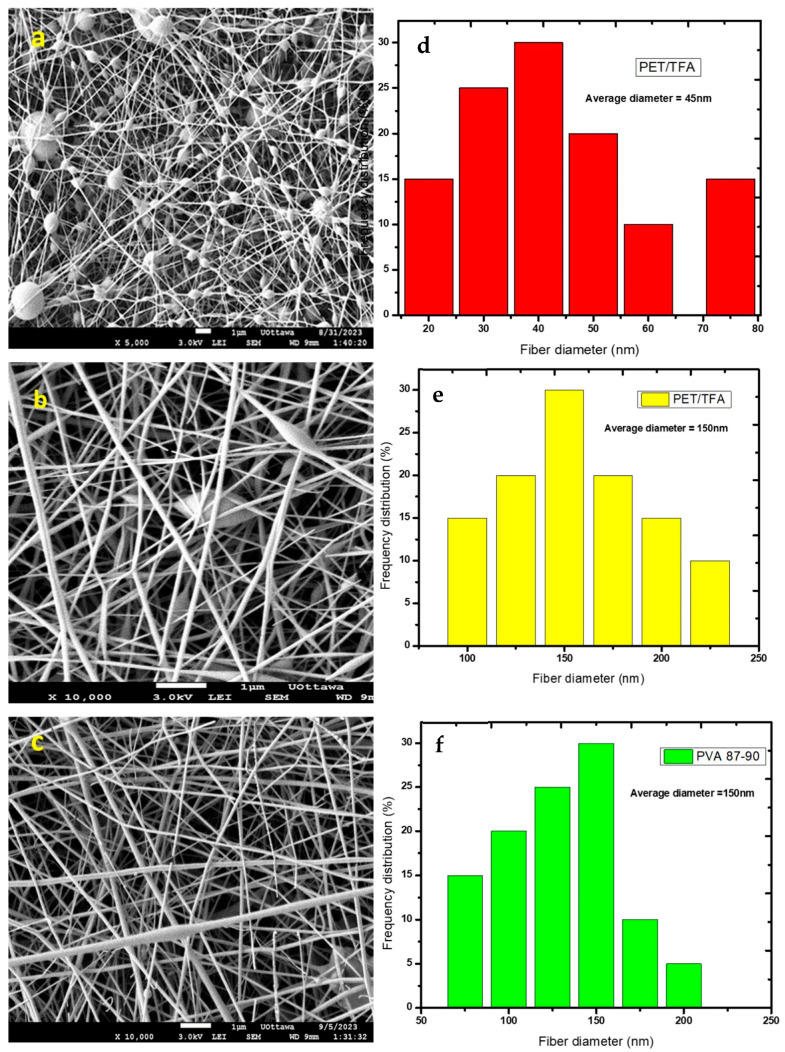
(**a**,**d**) PET TFA voltage 10 kV, (**b**,**e**) PVA WATER voltage 15 kV; (**c**,**f**) PVA WATER 18 kV at 15 cm tip-to-collector distance, 18 G, and 0.3 mL/h flowrate.

**Figure 4 polymers-16-03297-f004:**
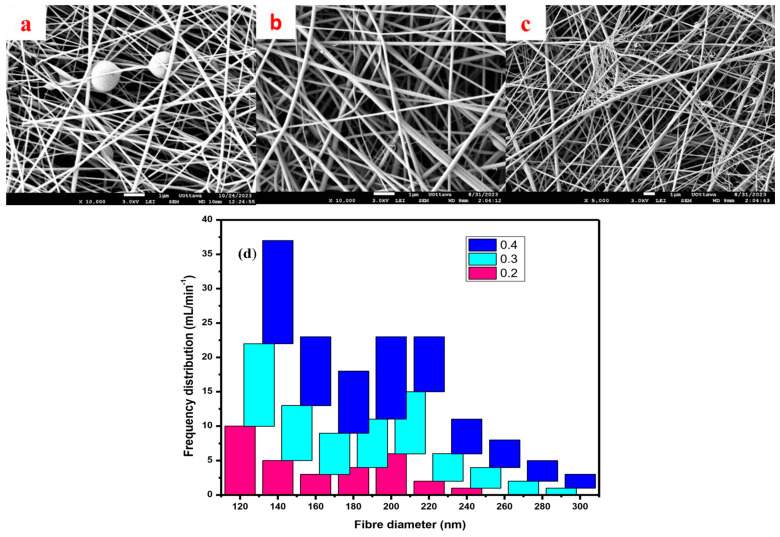
PVA/water, 15 cm tip-to-collector distance, 18 G, 18 kV: (**a**) flowrate = 0.2 mL/h, (**b**) flowrate = 0.3 mL/h, (**c**) flowrate = 0.4 mL/h; (**d**) frequency distribution.

**Figure 5 polymers-16-03297-f005:**
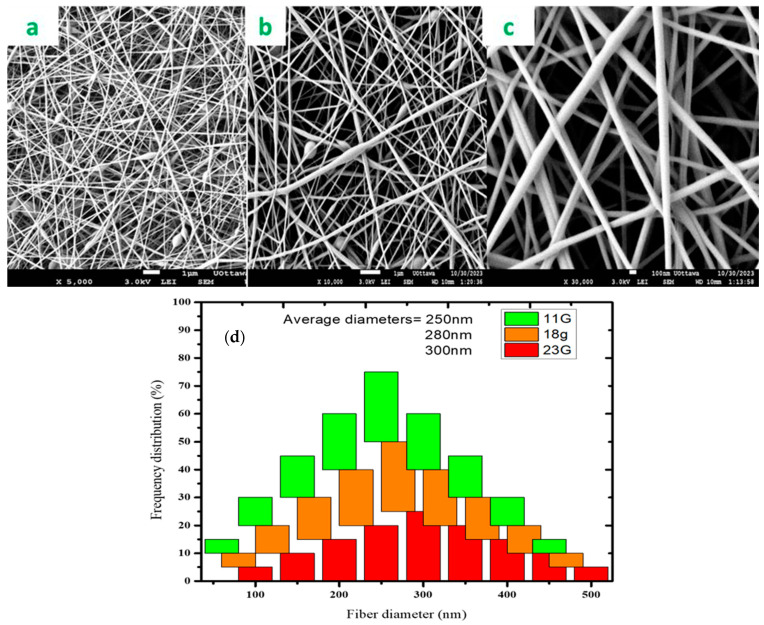
PVA/water and ethanol, 15 cm tip-to-collector distance, 0.3 mL/h flowrate, 18 kV: (**a**) needle = 23 G, (**b**) needle = 18 G; (**c**) needle = 11 G and fiber diameter frequency graph; (**d**) fiber diameter frequency graph.

**Figure 6 polymers-16-03297-f006:**
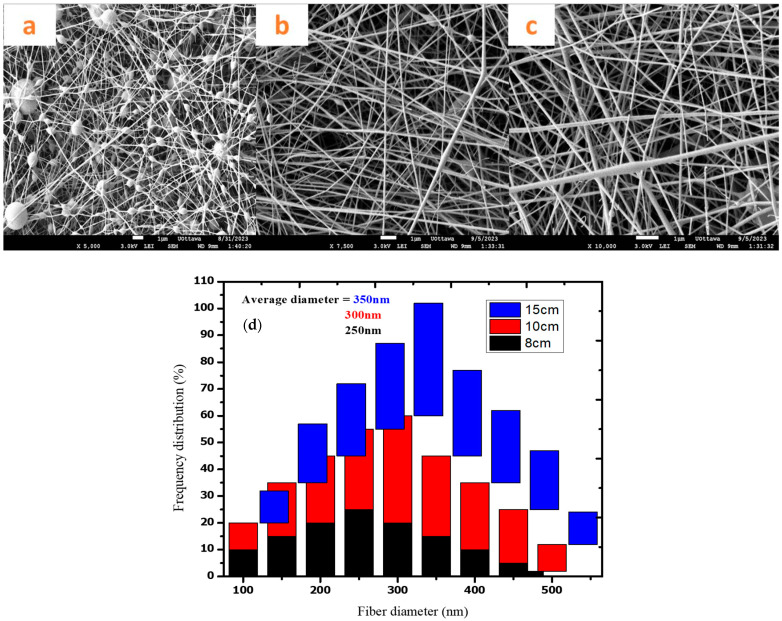
PVA/water, 18 G, 0.3 min/min flow rate, 18 kV: (**a**) distance = 8 cm, (**b**) distance = 10 cm; (**c**) distance = 15 cm and (**d**) frequency and fiber diameter graph.

**Figure 7 polymers-16-03297-f007:**
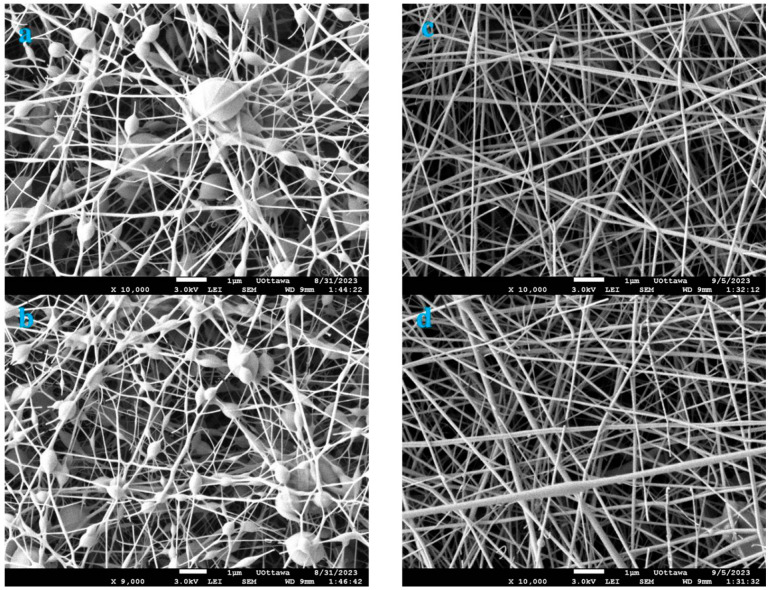
Solvent polymer effects: (**a**) PET TFA, (**b**) PET TFA/DCM, (**c**) PVA water and (**d**) PVA ETOH/water.

**Figure 8 polymers-16-03297-f008:**
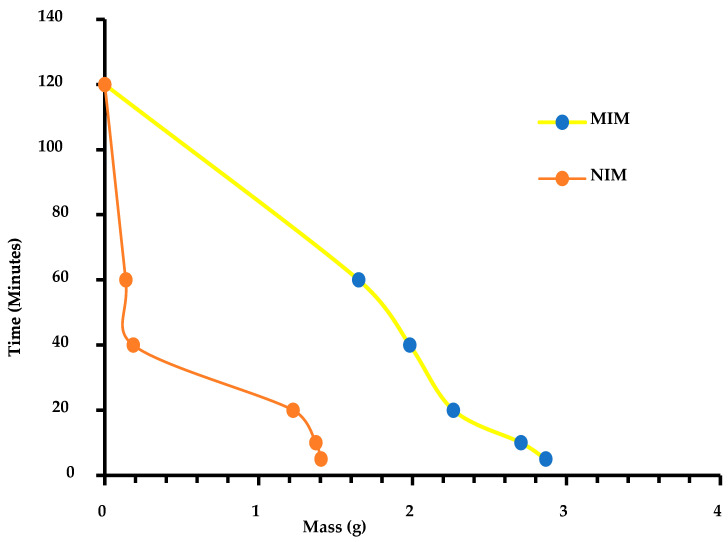
MIM and NIM swelling studies.

**Figure 9 polymers-16-03297-f009:**
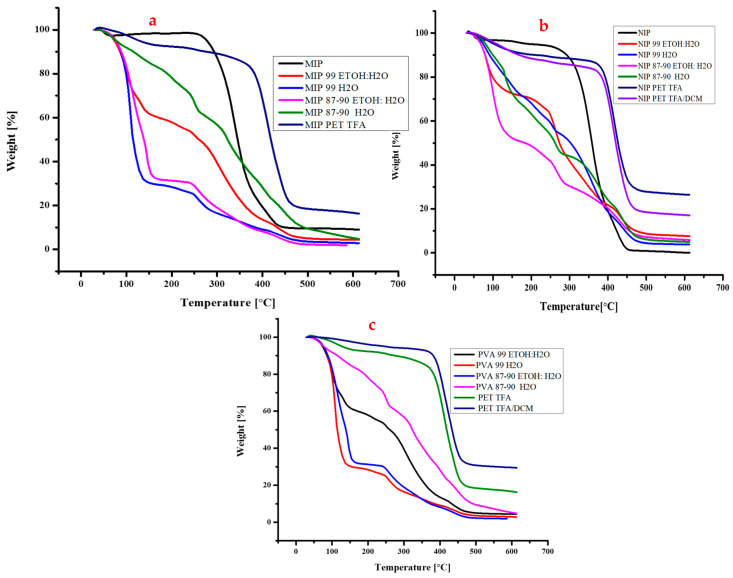
Thermal analysis of standard polymer materials, (**a**) MIP and (**b**) NIP additives to polymer solutions (**c**) polymer solutions.

**Figure 10 polymers-16-03297-f010:**
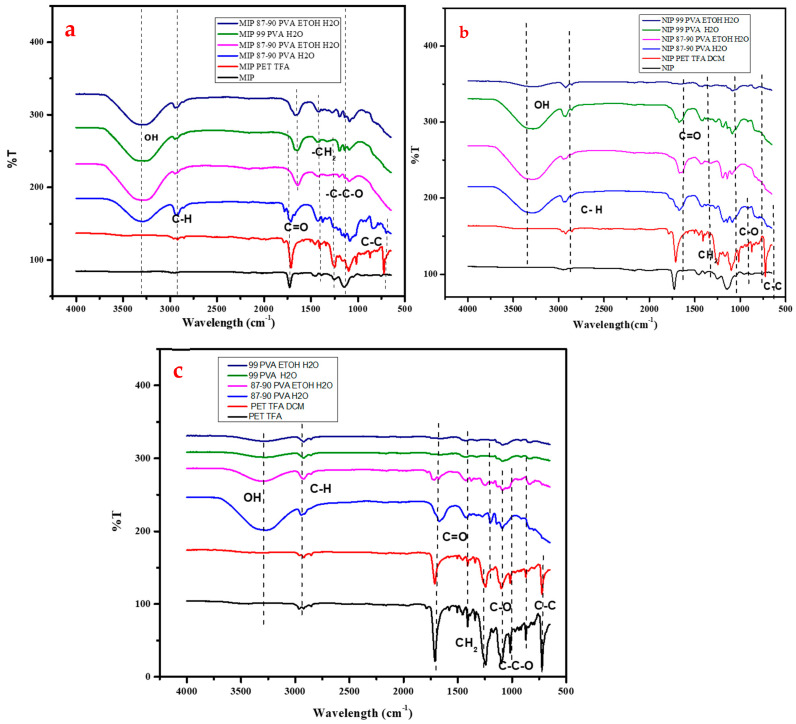
FTIR spectra of PVA, PET solution with (**a**) MIP and (**b**) NIP additives and (**c**) without.

**Figure 11 polymers-16-03297-f011:**
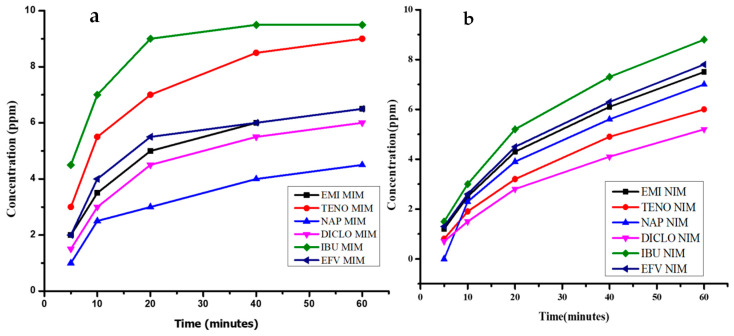
Adsorption capacity over time of MIP-PVA-based MIM (**a**) and NIM (**b**).

**Table 1 polymers-16-03297-t001:** Properties of polymers solutions.

Sample No.	Polymer	Polymer Ratio (%)	Solvent	Solvent Volume (mL)	Conductivity (µS/cm)	Viscosity (mPa·s)
1.	PET	100	TFA	40	33.6	12.7
2.		100	TFA and DCM	30:10	11.64	3.9
3.	PVA DH (87–90%)	100	Water	40	1054	165.5
4.		100	Water and ethanol	30:10	987	229.6
5.	PET MIP	98.6/1.4	TFA	40	70.5	35.2
6.		98.6/1.4	TFA and DCM	30:10	24.35	−2.7
7.	PVA DH (87–90%) MIP	98.6/1.4	Water	40	1190	154.9
8.		98.6/1.4	Water and ethanol	30:10	465	307.2
9.	PET NIP	98.6/1.4	TFA	40	72.9	18.3
10.		98.6/1.4	TFA and DCM	30:10	13.46	−0.7
11.	PVA DH (87–90%) NIP	98.6/1.4	Water	40	1082	167
12.		98.6/1.4	Water and ethanol	30:10	550	167.5

**Table 2 polymers-16-03297-t002:** PVA 99% polymer properties.

Sample No.	Polymer	Polymer Ratio (%)	Solvent	Solvent Volume (mL)	Conductivity (µS/cm)	Viscosity (mPa·s)
1.	PVA DH (99%)	100	water	40	839	2058.8
2.		100	Water and ethanol	30:10	425	249.8
3.	PVA DH (99%) MIP	98.6/1.4	Water	40	1386	2276.4
4.		98.6/1.4	Water and ethanol	30:10	666	586.6
5.	PVA DH (99%) NIP	98.6/1.4	Water	40	698	1316.2
6.		98.6/1.4	Water and ethanol	30:10	351	2314.8

## Data Availability

The original contributions presented in the study are included in the article/Appendix A; further inquiries can be directed to the corresponding author.

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
