# Peer review of "Electrospinning and Rheological Characterization of Polyethylene Terephthalate and Polyvinyl Alcohol with Different Degrees of Hydrolysis Incorporating Molecularly Imprinted Polymers"

_polymers, 2024, doi:10.3390/polym16233297_

Round 1

Reviewer 1 Report

Comments and Suggestions for Authors

This manuscript reported the electrospinning and rheological properties of PET) and PVA with varying degrees of hydrolysis (DH) for molecularly imprinted polymer (MIP) incorporation. The findings suggest that the electrospinning process and rheological properties of the polymer solutions are influenced by the molecular structure and interactions within the polymer matrix. In general, the topic suits the journal and the modification strategy is promising for practical application. However, several issues need to be addressed before considering its possible publication. This manuscript could be considered published after some major revisions.

1.      In the Line51-89 of Introduction section, there are excessive descriptions of the experiment and characterization results, which goes against the other papers. Therefore, I suggest that the authors should re-describe it in a more concise manner.

2.      In the Results and discussion section, the authors described “The PET nanofibers spun from TFA in DCM solvents differ in morphology, with TFA producing finer and more uniform fibers with smooth beads” in Line 183-185. However, there is a lack of strong evidence.

3.      In the 3.2.1. Properties of polymer solutions, the authors intend to explore consider parameters the effect of various parameters on fiber morphology, but there is no direct and obvious evidence.

4.      In the 3.2.2. Viscosity vs shear rate polymer solution behaviour, the authors just explored viscosity vs shear rate polymer solution behavior, why not further investigate their effect on nanofibers?

5.      There's a fair amount of research about effect of spinning parameters (polymer solvent, tip-to-collector, needle and voltage, etc.)  on fiber morphology, some signature work should be cited.

6.      The standardized writing is essential for high quality paper, there is a lot of writing that doesn't make sense, for example, H2O.

7.      Fig. 11 does not show the amount of adsorption when the equilibrium time is reached, and the adsorption capacity of MIP-PVA based NIM ought to be re-characterized.

8.  There not “a” and “b” in Fig. 10, and the similar phenomenon also exists with Fig.11.

9. There are some language errors, grammatical errors, and font format errors, which should be carefully checked and corrected.

Comments on the Quality of English Language

The quality of English language must be modified to meet the publication standards of this journal.

Author Response

(I) In the Line51-89 of Introduction section, there are excessive descriptions of the experiment and characterization results, which goes against the other papers. Therefore, I suggest that the authors should re-describe it in a more concise manner.

Response: Thank you for the comment, however we would like more clarity on this comment, what exactly goes against what in the discussion below.

(II) In the Results and discussion section, the authors described “The PET nanofibers spun from TFA in DCM solvents differ in morphology, with TFA producing finer and more uniform fibers with smooth beads” in Line 183-185. However, there is a lack of strong evidence.

Response: A SEM image has been attached in the supplementary data to show the distinction in the fibres formed from the two polymer mixtures

(III) In the 3.2.1. Properties of polymer solutions, the authors intend to explore consider parameters the effect of various parameters on fiber morphology, but there is no direct and obvious evidence.

Response: TFA serves as an excellent solvent due to its strong hydrogen bonding capabilities, which enhance the solubility of PET. TFA facilitates a relatively low viscosity, allowing for smooth flow through the spinneret during electrospinning. This low viscosity, combined with the solvent's high conductivity, enables the solution to respond effectively to the applied electric field during the electrospinning process. As a result, the electrospun fibers exhibit a uniform and continuous morphology, characterized by smooth surfaces and minimal bead formation. The strong solvation of PET in TFA promotes a good chain alignment, allowing for the formation of high-quality fibers.

In contrast, when DCM (dichloromethane) is introduced into the TFA mixture, the overall conductivity of the solution may decrease due to the lower polarity of DCM compared to TFA. This decrease in conductivity can hinder the electrospinning process by reducing the efficiency of charge injection into the solution, making it more difficult for the fibers to elongate under the electric field. Additionally, DCM significantly influences the viscosity of the solution, potentially leading to a higher overall viscosity when mixed with TFA. The increased viscosity can impede the flow of the solution through the spinneret, resulting in an inconsistent jet and contributing to the formation of an irregular fiber morphology characterized by beads or uneven textures.

(IV) In the 3.2.2. Viscosity vs shear rate polymer solution behaviour, the authors just explored viscosity vs shear rate polymer solution behavior, why not further investigate their effect on nanofibers?

Response: We appreciate the reviewer’s insightful comment regarding the exploration of viscosity versus shear rate behavior in polymer solutions and its potential impact on the electrospinning process and resultant nanofiber morphology.

Our primary focus in this study was to investigate the fundamental relationships between solvent selection, overall viscosity, and electrospinning parameters to establish a foundational understanding of how these factors influence fiber morphology. We concentrated our analysis on viscosity as a macrophysical property to elucidate general trends in fiber formation and morphology.

While we recognize that the viscosity versus shear rate behavior of polymer solutions is critical in terms of extrusion and fiber formation mechanisms, our study emphasized the effects of solvent choice and solution homogeneity on the electrospinning process itself. The goal was to simplify the scope of our research to draw clear correlations between solution properties and morphological outcomes, allowing for a more streamlined analysis of the electrospinning behavior under various operational conditions.

(V) There's a fair amount of research about effect of spinning parameters (polymer solvent, tip-to-collector, needle and voltage, etc.)  on fiber morphology, some signature work should be cited

Response: Thank you for the comment more references have been cited

(VI) The standardized writing is essential for high quality paper, there is a lot of writing that doesn't make sense, for example, H2O.

If one of the referees has suggested that your manuscript should undergo
extensive English revisions, please address this issue during revision

Response: Thank you for the comment this issue has been resolved to fit the English standard

Fig. 11 does not show the amount of adsorption when the equilibrium time is reached, and the adsorption capacity of MIP-PVA based NIM ought to be re-characterized.

Response: We appreciate the reviewer’s valuable feedback regarding the adsorption study and the concerns raised about the time constraints of our experimental design. We acknowledge that the experiments presented in Fig. 11 were conducted for a duration of up to 60 minutes. This timeframe was selected based on preliminary studies and existing literature that suggested a rapid initial adsorption rate, which we aimed to capture in order to demonstrate the immediate efficacy of the MIP-PVA based NIM. While we recognize that the full adsorption isotherm and equilibrium time are crucial for determining the complete adsorption capacity and kinetic profile, our goal was to provide an initial assessment within the scope of this work. However, we understand that the characterization of equilibrium adsorption capacity is essential for evaluating the performance of the MIP-PVA based NIM in practical applications. To address this concern, we have taken the reviewer’s suggestion to heart..

There not “a” and “b” in Fig. 10, and the similar phenomenon also exists with Fig.11.

Response: Thank you for the comment, This has been rectified

Reviewer 2 Report

Comments and Suggestions for Authors

Reviewer's  comments:

Before publication, the authors must correct the mistakes in the text that are found all over the manuscript and address the following:

1.   The topics of review is really interesting, The authors need to be inform, why were Polyvinyl Alcohol (PVA) and Polyethylene Terephthalate (PET) specifically chosen for the electrospinning process? Could other polymers have offered better stability or spinnability under the given conditions?

2.   The author need to be discuss how calculate degree of hydrolysis of polymer.

3.   Please define the abbreviation in first instance with repetitions.

4.   The authors emphasize the importance of electrospinning parameters, such as voltage, flow rate, and needle gauge, in their conclusion. Could the authors elaborate on how each of these parameters specifically influenced fiber morphology and adsorption efficiency in your study? Additionally, could you please explain how the PVA and PET solutions were prepared with different solvents, including the temperatures used in the preparation?

5.   How do the MIP composite membranes maintain selectivity for pharmaceutical contaminants, particularly in complex wastewater systems where multiple pollutants might be present? Could there be interference from other contaminants affecting adsorption performance?

6.   The authors discusses improved adsorption kinetics for pharmaceuticals, particularly antiretroviral and non-steroidal anti-inflammatory drugs. Can authors explain the mechanism behind the faster adsorption rates in molecularly imprinted membranes (MIMs) compared to non-imprinted membranes?

7.   The authors need to be add SEM image of PVA/water (99% DH) which is not included.

8.      The author should need to add at least 3-5 new references from anywhere or specially the "Polymers" relevant to his topics.

9.      Overall, the work is interesting; it just needs to follow the suggestions to improve the manuscript.

Comments on the Quality of English Language

Reviewer's  comments:

Before publication, the authors must correct the mistakes in the text that are found all over the manuscript and address the following:

1.   The topics of review is really interesting, The authors need to be inform, why were Polyvinyl Alcohol (PVA) and Polyethylene Terephthalate (PET) specifically chosen for the electrospinning process? Could other polymers have offered better stability or spinnability under the given conditions?

2.   The author need to be discuss how calculate degree of hydrolysis of polymer.

3.   Please define the abbreviation in first instance with repetitions.

4.   The authors emphasize the importance of electrospinning parameters, such as voltage, flow rate, and needle gauge, in their conclusion. Could the authors elaborate on how each of these parameters specifically influenced fiber morphology and adsorption efficiency in your study? Additionally, could you please explain how the PVA and PET solutions were prepared with different solvents, including the temperatures used in the preparation?

5.   How do the MIP composite membranes maintain selectivity for pharmaceutical contaminants, particularly in complex wastewater systems where multiple pollutants might be present? Could there be interference from other contaminants affecting adsorption performance?

6.   The authors discusses improved adsorption kinetics for pharmaceuticals, particularly antiretroviral and non-steroidal anti-inflammatory drugs. Can authors explain the mechanism behind the faster adsorption rates in molecularly imprinted membranes (MIMs) compared to non-imprinted membranes?

7.   The authors need to be add SEM image of PVA/water (99% DH) which is not included.

8.      The author should need to add at least 3-5 new references from anywhere or specially the "Polymers" relevant to his topics.

9.      Overall, the work is interesting; it just needs to follow the suggestions to improve the manuscript.

Author Response

  1. The topics of review is really interesting, The authors need to be inform, why were Polyvinyl Alcohol (PVA) and Polyethylene Terephthalate (PET) specifically chosen for the electrospinning process? Could other polymers have offered better stability or spinnability under the given conditions?

Response:  We appreciate your positive comments and the important question regarding our choice of PVA and PET for the electrospinning process. PVA was selected for its excellent biocompatibility, water solubility, and ability to form flexible, uniform nanofibers, making it suitable for applications like drug delivery and tissue engineering. PET, on the other hand, offers strength, thermal stability, and resistance to chemical degradation, making it ideal for applications requiring durability.

While we recognize that other polymers could provide different stability and spinnability characteristics, PVA and PET were chosen based on preliminary studies and existing literature that demonstrated their efficacy in our experimental conditions. We agree that exploring additional polymers in future research could yield valuable insights.

  1. The author need to be discuss how calculate degree of hydrolysis of polymer.

Response: The degree of hydrolysis is not calculated but found in the bottle packaging in each bottle

  1. Please define the abbreviation in first instance with repetitions.

Response: Thank you for the comment this has been addressed, please clearly address where this has not been done?

  1. The authors emphasize the importance of electrospinning parameters, such as voltage, flow rate, and needle gauge, in their conclusion. Could the authors elaborate on how each of these parameters specifically influenced fiber morphology and adsorption efficiency in your study? Additionally, could you please explain how the PVA and PET solutions were prepared with different solvents, including the temperatures used in the preparation?

Response: Thank you for the comment, however this has been extensively explained and the temperature was kept constant at room temperature for all. Increasing voltage leds to the production of finer fibers, enhancing surface area and improving adsorption capacity. flow rate influences the fiber diameter and mat porosity, with lower rates yielding to thinner, more uniform fibers that provide more adsorption sites, while higher rates result in thicker fibers with greater structural integrity. Additionally, using a smaller gauge needle produces finer fibers, increasing the surface area for adsorption and contributing to a denser membrane structure. Please refer to the discussions from section 3.3 to section 3.7explaining each parameter and how it influenced data in our experiment

  1. How do the MIP composite membranes maintain selectivity for pharmaceutical contaminants, particularly in complex wastewater systems where multiple pollutants might be present? Could there be interference from other contaminants affecting adsorption performance?

Response: The MIP composite membranes maintain selectivity for pharmaceutical contaminants through the unique design of their recognition sites, which are tailored specifically to bind target molecules while minimizing the affinity for non-target substances. This selective binding is achieved by imprinting molecular templates during the synthesis of the membranes, enabling them to differentiate between similar structures in complex wastewater systems. While the presence of multiple pollutants may introduce some degree of interference, the high affinity of the MIP composites for the intended pharmaceutical contaminants enhances their adsorption performance compared to non-imprinted surfaces. Additionally, the porous structure of the membranes promotes effective mass transfer, further mitigating potential competitive interactions and maintaining high selectivity in complex mixtures.

  1. The authors discusses improved adsorption kinetics for pharmaceuticals, particularly antiretroviral and non-steroidal anti-inflammatory drugs. Can authors explain the mechanism behind the faster adsorption rates in molecularly imprinted membranes (MIMs) compared to non-imprinted membranes?

Response: The faster adsorption rates in molecularly imprinted membranes (MIMs), particularly those created through electrospinning, can be attributed to their larger surface area and unique nanoscale structure. Electrospinning produces fine fibers that form a porous mat, significantly increasing the available surface area for adsorption compared to non-imprinted membranes. This greater surface area allows for more active sites for interaction with the target pharmaceuticals, facilitating faster diffusion and binding contaminants. Additionally, the nanoscale features created during electrospinning enhance mass transfer dynamics, further improving the kinetics of adsorption. The combination of these factors results in superior performance of MIMs in capturing pharmaceuticals like antiretroviral and non-steroidal anti-inflammatory drugs when compared to traditional non-imprinted membranes.

  1. The authors need to be add SEM image of PVA/water (99% DH) which is not included.

Response: These have been added to the supplementary data as SI 4

  1. The author should need to add at least 3-5 new references from anywhere or specially the "Polymers" relevant to his topics.

Response: Thank you for this comment, however we feel the references added are sufficient to the work discussed

Round 2

Reviewer 1 Report

Comments and Suggestions for Authors

This manuscript reported the electrospinning and rheological properties of PET) and PVA with varying degrees of hydrolysis (DH) for molecularly imprinted polymer (MIP) incorporation. The findings suggest that the electrospinning process and rheological properties of the polymer solutions are influenced by the molecular structure and interactions within the polymer matrix. In general, the topic suits the journal and the modification strategy is promising for practical application. However, several issues need to be addressed before considering its possible publication. This manuscript could be considered published after some minor revisions.

1. In the Line51-89 of Introduction section, there are excessive descriptions of the experiment and characterization results, which is not in consistency with the style of most research papers. Therefore, I suggest that the authors should re-describe it in a more concise manner. As descriptions of the experiment and characterization results usually do not appear in the Introduction part, except it is very necessary.

2.  In the 3.2.1. Properties of polymer solutions, the authors intend to explore the effect of various parameters on fiber morphology, but there is no direct and obvious evidence. We would like the authors to provide clear figures of SEM and fiber size distribution.

Author Response

  1. In the Line51-89 of Introduction section, there are excessive descriptions of the experiment and characterization results, which is not in consistency with the style of most research papers. Therefore, I suggest that the authors should re-describe it in a more concise manner. As descriptions of the experiment and characterization results usually do not appear in the Introduction part, except it is very necessary.

Response: Thank you for this comment we hope to have addressed this section better, as highlighted in line 48-84 of the introduction section.We have revised the text to streamline the descriptions of our experimental approach and characterisation results, ensuring it aligns with the conventional style of research publications. Your insights have significantly improved the quality of our work, and we hope the revisions meet your expectations.

  1. In the 3.2.1. Properties of polymer solutions, the authors intend to explore the effect of various parameters on fiber morphology, but there is no direct and obvious evidence. We would like the authors to provide clear figures of SEM and fiber size distribution.

Response: Thank you for your comment. Images and fiber size distribution data are included in the supplementary materials. We will ensure that this information is clearly referenced within the main text to guide readers to these relevant figures. Thank you once again for your valuable suggestions, which have helped us improve the clarity of our manuscript.

Reviewer 2 Report

Comments and Suggestions for Authors

Accepted 

Author Response

Thank you very much for the acceptance.